# Regulation of neuronal axon specification by glia-neuron gap junctions in *C. elegans*

**Lingfeng Meng[1,2,3], Albert Zhang[1,2,3], Yishi Jin[4,5], Dong Yan[1,2,3]\***

[1]Department of Molecular Genetics and Microbiology, Duke University Medical Center, Durham, United States; [2]Department of Neurobiology, Duke University Medical Center, Durham, United States; [3]Duke Institute for Brain Sciences, Duke Medical Center, Durham , United States; [4]Neurobiology Section, Division of Biological Sciences, Howard Hughes Medical Institute, University of California, San Diego, United States; [5]Department of Cellular and Molecular Medicine, School of Medicine, University of California, San Diego, United States

**Abstract** Axon specification is a critical step in neuronal development, and the function of glial cells in this process is not fully understood. Here, we show that *C. elegans* GLR glial cells regulate axon specification of their nearby GABAergic RME neurons through GLR-RME gap junctions. Disruption of GLR-RME gap junctions causes misaccumulation of axonal markers in non-axonal neurites of RME neurons and converts microtubules in those neurites to form an axon-like assembly. We further uncover that GLR-RME gap junctions regulate RME axon specification through activation of the CDK-5 pathway in a calcium-dependent manner, involving a calpain *clp-4*. Therefore, our study reveals the function of glia-neuron gap junctions in neuronal axon specification and shows that calcium originated from glial cells can regulate neuronal intracellular pathways through gap junctions.

**\*For correspondence:** dong.yan@duke.edu

**Competing interests:** The authors declare that no competing interests exist.

## Introduction

Glial cells play important roles in neural circuit formation during development, synaptic plasticity and injury repair in adulthood (*Bosse, 2012*; *Clarke and Barres, 2013*; *Nave and Trapp, 2008*). In mammals, they are involved in many steps of brain development: in embryos, radial glial cells serve as stem cells (*Doetsch et al., 1999*; *Noctor et al., 2001*) and act as scaffolds for neuronal migration and axon growth (*Chotard and Salecker, 2004*); in newborn animals, astrocytes regulate synaptogenesis through contact-mediated or secreted signals, such as glypicans, thrombospondins, hevin and SPARC (*Allen et al., 2012*; *Christopherson et al., 2005*; *Elmariah et al., 2005*; *Hama et al., 2004*; *Hughes et al., 2010*; *Kucukdereli et al., 2011*; *Pfrieger and Barres, 1997*; *Ullian et al., 2001*); and in mature neurons, astrocytes and microglia regulate refinement of neural circuits (*Chung et al., 2013*; *Schafer et al., 2012*; *Stevens et al., 2007*). Similar studies using invertebrate systems have also added to our knowledge of glial cells in neural development. Studies in *Drosophila* have identified important roles of glial cells in neurogenesis, axon development and pruning, and synapse formation and elimination (*Corty and Freeman, 2013*; *Coutinho-Budd and Freeman, 2013*). *C. elegans* has 56 glial cells, which consist of 46 neuroepithelial glia that ensheath sensory dendrites, four neuroepithelial glia that envelop the nerve ring, and six GLR cells that reside in the inner nerve ring (*Oikonomou and Shaham, 2011*). *C. elegans* neuroepithelial glial cells are known to regulate axon growth, dendrite development, and synapse specification through secreted and contact-mediated signals (*Colon-Ramos et al., 2007*; *Heiman and Shaham, 2009*; *Shao et al., 2013*; *Yoshimura et al., 2008*), but the function of GLR cells in neuronal development is still unknown.

Neurons are polarized cells with two functionally distinct domains: axons and dendrites. Most neurons have multiple dendrites to receive stimulation from upstream cells and a single axon to pass signals to downstream targets, and this general morphology is essential for them to integrate and transmit information in the brain. While glial cells are involved in regulating many aspects of neuronal development and function, whether and how glial cells affect axon specification remains largely unknown.

During mammalian brain development, transient gap junctions between neurons are formed at many regions to regulate synaptogenesis, neuronal differentiation, neuronal migration, and neural circuit formation (*Belousov and Fontes, 2013*; *Connors and Long, 2004*; *Li et al., 2012*; *Yu et al., 2012*). In invertebrates, transient neuronal gap junctions regulate the formation of chemical synapses in leeches (*Todd et al., 2010*) and are required for the asymmetrical development of sensory neurons in *C. elegans* (*Chuang et al., 2007*). Electron microscopy and dye-filling studies have shown that neurons and glial cells also form transient gap junctions in the developing brain (*Cuadras et al., 1985*; *Nadarajah et al., 1997*; *Nadarajah and Parnavelas, 1999*; *Nadarajah et al., 1996*; *Peinado et al., 1993*; *Shivers et al., 1988*; *Sinues Porta et al., 1988*). Further electrophysiology analysis confirmed that those neuron-glia gap junctions are likely to be functional (*Alvarez-Maubecin et al., 2000*; *Froes et al., 1999*; *Nedergaard, 1994*; *Nedergaard and Goldman, 1993*); however, it remains unknown whether they play any roles in neuronal development.

Here, we show that gap junctions between *C. elegans* GLR glial cells and GABAergic RME neurons are required for RME axon specification. In an unbiased genetic screen, we uncovered a loss-of-function mutant of *unc-1* with defects in RME axon specification. UNC-1 is a conserved stomatin protein that plays important roles in regulating gap junctions (*Chen et al., 2007*; *Rajaram et al., 1998*). We show that *unc-1* functions in both GLR cells and RME neurons to regulate RME axon specification. Mutating all RME-expressed gap junction channels causes the same extent of axon specification defects as seen in *unc-1(lf)* mutants, supporting the conclusion that *unc-1* regulates RME axon specification by modulating GLR-RME gap junctions. Using a combination of in vivo imaging, biochemistry, and genetic studies, we demonstrate that GLR-RME gap junctions can regulate microtubule assembly in RME neurites through activation of the CDK-5 pathway in a calcium-dependent manner. As the function of the CDK-5 pathway in regulating microtubules is highly conserved from nematodes to mammals, a similar mechanism may also be used in other organisms.

## Results

### *unc-1* is required for RME axon specification

To study neural circuit formation in vivo, we used the *C. elegans* RME circuit as a model (*Meng et al., 2015*). The circuit contains four GABAergic motor neurons, RMED, RMEV, RMEL, and RMER, that reside at the nerve ring region to control head movement (*Song et al., 2010*; *White et al., 1986*). The axonal processes of RME neurons form a ring-shaped bundle, and RME D/V extend two neurites along the dorsal/ventral cords (*Figure 1A and B*). In adult control animals, presynaptic markers such as synaptobrevin (SNB-1) exclusively accumulate in axonal boutons at the end of the ring-shaped axonal process (*Figure 1C and D*)(*White et al., 1986*). In a genetic screen for RME neuronal development, we isolated a mutant, *ju1057*, with SNB-1::GFP puncta present along RME axonal processes and D/V neurites in addition to those in normal axonal boutons (*Figure 1C and D*). Careful analyses of *ju1057* phenotypes showed that *ju1057* did not affect the overall expression level of SNB-1::GFP but significantly changed the distribution of SNB-1::GFP. In *ju1057 animals*, a higher fraction of SNB-1::GFP signal was observed in RME D/V neurites (47%) than in axonal boutons (40%), while in control animals 8% of SNB-1::GFP signals were observed in D/V neurites and 67% of them at axonal boutons (*Figure 1—figure supplement 1A*). *ju1057* animals displayed kinker uncoordinated (*unc*) phenotypes. Based on SNB-1::GFP phenotypes, *ju1057* was mapped to the left arm of the chromosome X, a region containing the classic kinker *unc* gene *unc-1*. *ju1057* failed to complement two *unc-1* loss-of-function alleles, *e580* and *e2522*; and *e580* and *e2522* animals displayed SNB-1::GFP mislocalization defects in both axonal processes and D/V neurites resembling those in *ju1057* animals (*Figure 1D*). Sequencing results showed that *ju1057* changed a conserved alanine to valine (A248V) in the C-terminal domain of UNC-1 (*Figure 1—figure supplement 1B*). In addition, expression of *unc-1* cDNA in all tissues using the *Pdpy-30* promoter

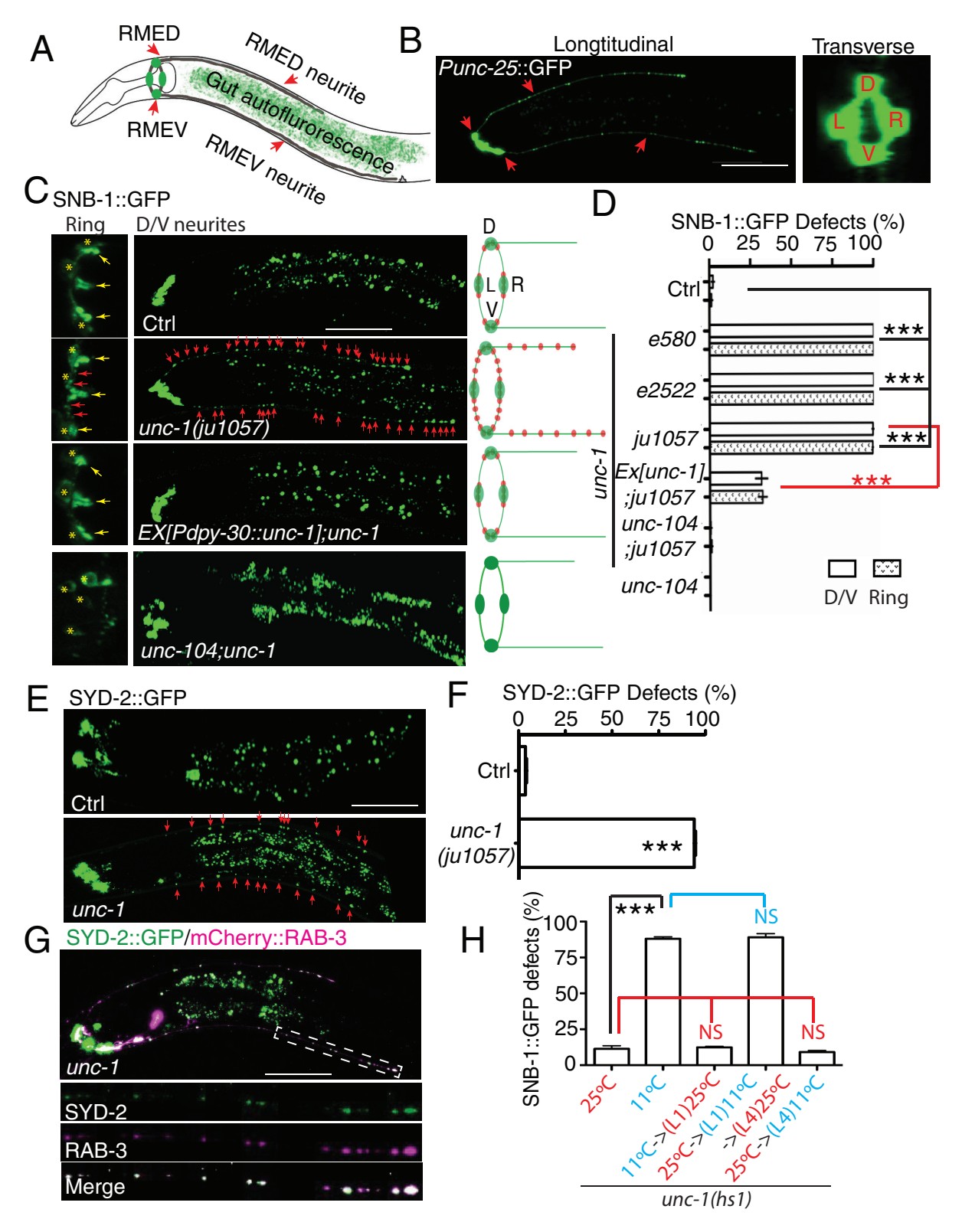

**Figure 1.** *unc-1* is required for RME axon specification. (**A**) A cartoon picture and (**B**) GFP images show the morphology of RME neurons. Cell bodies of four RME neurons, labeled as D (RME dorsal), V (RME ventral), L (RME left) and R (RME right), form a four-fold symmetry in the head region. RMED and RMEV each sends out a posterior neurite along the dorsal/ventral cords, respectively. Red arrows highlight RMED/V cell bodies and neurites. Gut autofluorescence from the intestine is commonly observed between RMED/V neurites. (**C**) Confocal images and schematic diagrams show the distribution of

*Figure 1 continued on next page*

*Figure 1 continued*

SNB-1::GFP puncta in control and mutant animals. Yellow stars mark the cell bodies of RME neurons, and yellow arrowheads point to axonal boutons. Red arrowheads mark the mis-accumulated SNB-1::GFP puncta. Red dots in diagrams represent SNB-1::GFP puncta. (D) Quantification of the percentage of animals with SNB-1::GFP puncta in RME D/V neurites and along the axonal processes. (E) Confocal images and schematic diagrams of control and *unc-1(lf)* animals expressing SYD-2::GFP in RME neurons. (F) Quantification of percentage of animals with SYD-2::GFP puncta in RMED/V neurites. (G) SYD-2::GFP is co-localized with mCherry:: RAB-3 in RMED/V neurites of *unc-1(lf)* animals. (H) *unc-1* regulates the establishment of RME axon specification during development. (See details of the temperature-shift experiments in the method section). Each experiment was performed with N > 200 animals at least three times. For transgenic animals, the results shown here are generated from at least three independent lines. Data are shown as mean ± SD. Student t-test, \*\*\*p<0.001, NS: no significant difference. Scale bar: 20 μm. Details of strain and plasmid information in all Figures are listed in the *Supplementary file 1.*

The following figure supplements are available for figure 1:

**Figure supplement 1.** *ju1057* is a loss-of-function allele of *unc-1.*

**Figure supplement 2.** *unc-1(lf)* affects RME axon specification.

rescued the SNB-1::GFP defects of *ju1057* animals (*Figure 1C and D*). These lines of evidence support that *ju1057* is a loss-of-function allele of *unc-1.*

Our genetic screen was carried out in a *unc-30(ju32)* mutant background to eliminate the expression of *Punc-25::SNB-1::GFP* in D-type motor neurons in the dorsal and ventral cords. To verify that the defects we observed did not depend on *unc-30(ju32)*, we examined *unc-1* animals without *unc-30(ju32)*. As shown in *Figure 1—figure supplement 2A*, we found mis-accumulated SNB-1::GFP puncta along the axonal process in all *unc-1(lf)* animals we examined (105/105), similar to those seen in *unc-30(ju32)* background. RME D/V neurites grow along the dorsal and ventral cords together with D-type motor neurons, making it difficult to observe RME D/V phenotypes. However, in about 2–3% of animals, RME D neurites grow out of the dorsal cord. After screening over 500 *unc-1(lf)* animals, we were able to recover 14 animals with RME D neurites outside of the dorsal cord, and all of them (14/14) displayed SNB-1::GFP puncta in RME D neurites (*Figure 1—figure supplement 2A*).

To further illustrate the degree of RME phenotypes, we quantified the number of SNB-1::GFP puncta in RME D neurites and found that all three *unc-1(lf)* alleles had a similar number of SNB-1::GFP puncta (25–30 puncta, *Figure 1—figure supplement 2B*). The axonal localization of SNB-1::GFP relies on the kinesin UNC-104/KIF1A (*Hall and Hedgecock, 1991*). Loss-of-function of *unc-104* completely eliminated SNB-1::GFP signals in RME axonal processes and D/V neurites of *unc-1(lf)* animals (*Figure 1C and D*), supporting that the additional SNB-1::GFP puncta in *unc-1(lf)* animals are likely associated with presynaptic vesicles. In our analyses of SNB-1::GFP phenotypes, we noticed that the defects in RME axonal processes and D/V neurites displayed an all-or-none pattern such that any animals with the D/V neurite defects had SNB-1::GFP puncta along axonal processes, while animals without SNB-1::GFP puncta in D/V neurites did not show defects in axonal processes. This observation suggests that the defects in RME axonal processes and in D/V neurites are likely two aspects of one phenotype. Therefore, we decided to focus the rest of our analyses on RME D/V neurites. To further confirm *unc-1(lf)* phenotypes, we tested the distribution of other axonal markers, the presynaptic active zone protein SYD-2 (*Yeh et al., 2005*) and synaptic vesicle binding protein RAB-3. Consistent with the observation using the SNB-1::GFP marker, *unc-1(lf)* caused misaccumulation of SYD-2::GFP and mCherry::RAB-3, and SYD-2::GFP was also co-localized with mCherry::RAB-3 in RME D/V neurites (*Figure 1E,F and G*).

The presence of these axonal markers in D/V neurites suggests that mutation of *unc-1* disrupts certain aspects of axon specification in RME neurons. *unc-1(lf)* phenotypes could be due to defects in either the establishment of axon specification or the maintenance of neuronal polarity. To distinguish between these two possibilities, we took advantage of a cold-sensitive *unc-1(hs1, lf)* allele, that has strong *unc-1(lf)* locomotion phenotypes at 11°C and reverse to almost wild-type at 25°C (*Hecht et al., 1996*). At 25°C, less than 10% of *unc-1(hs1)* animals had RME defects, while more than 85% of animals displayed RME defects at 11°C (*Figure 1H*). Next, we examined the effect of temperature-shifts on *unc-1(hs1)* phenotypes. As shown in *Figure 1H*, growing animals at the restrictive temperature (11°C) before the L1 stage or after the L4 stage showed the same degree of RME

defects as those cultured at the permissive temperature (25°C), but growing larval-stage animals (L1-middle L4) at the restrictive temperature (11°C) was sufficient to cause RME defects to the same extent as those cultured at the restrictive temperature (11°C). These results support the conclusion that *unc-1* functions in the larval stages, the same time period of RME D/V neurite development (*Meng et al., 2015*), to regulate the establishment of RME axon specification.

## GLR-RME gap junctions are required for RME axon specification

*unc-1* encodes a stomatin protein and is widely expressed in almost all neurons and many other cells (*Chen et al., 2007*; *Rajaram et al., 1998*). To understand how *unc-1* regulates RME axon specification, we carried out tissue-specific rescue experiments. To our surprise, although expressing *unc-1 (cDNA)* in all tissues (*Pdpy-30:: unc-1(cDNA)*) rescued *unc-1(lf)* phenotypes, expression of *unc-1* cDNA in all neurons (*Punc-33:: unc-1(cDNA)*) or in RME neurons (*Punc-25:: unc-1(cDNA)*) did not rescue the RME defects in *unc-1(lf)* animals (*Figure 2A*), suggesting that *unc-1* functions either outside of the nervous system or in both neurons and non-neuronal cells to regulate RME axon specification.

Electrophysiology and genetic studies have shown that in *C. elegans*, *unc-1* plays important functions in regulating gap junctions, and *unc-1(lf)* blocked the electrical current mediated by gap junctions in muscle (*Chen et al., 2007*). RME neurons form gap junctions with themselves, several nearby neurons, and GLR cells (*White et al., 1986*). Given that neuronal expression of *unc-1* cDNA did not rescue *unc-1(lf)* phenotypes, we questioned whether the expression of *unc-1* in GLR cells was required for RME axon specification. We expressed *unc-1* cDNA in GLR cells using four different promoters (*Pgly-18*, *Pnep-2*, *Pegl-6,* and *Plet-2*). In addition to GLR cells, *Pgly-18*, *Plet-2,* and *Pegl-6* promoters could drive expression in head muscle and some neurons as well, but *Pnep-2* showed restricted expression in GLR cells (*Figure 2—figure supplement 1*). As shown in *Figure 2A*, expression of *unc-1* cDNA in GLR cells did not restore the distribution of SNB-1::GFP in *unc-1(lf)* animals, but co-expression of *unc-1* cDNA in both RME neurons and GLR cells using any one of the four promoters significantly rescued *unc-1(lf)* phenotypes (*Figure 2—figure supplement 1*). As controls, expression of *unc-1(cDNA)* in muscle (*Pmyo-3:: unc-1(cDNA)*), in both muscle and RME neurons (*Pmyo-3:: unc-1(cDNA)+ Punc-25:: unc-1(cDNA)*), or in both muscle and GLR cells (*Pmyo-3:: unc-1 (cDNA)+ Pgly-18:: unc-1(cDNA)*) did not show any rescue ability (*Figure 2A*). To further confirm that the expression of *unc-1* in both RME neurons and GLR cells was required for RME axon specification, we carried out mosaic analyses using a transgene (*yadEx644*) expressing *unc-1(cDNA)* in all tissues (*Pdpy-30:: unc-1(cDNA)*), a nuclear marker *sur-5::mCherry* in all tissues (*Psur-5:: sur-5::mCherry*), and mCherry in RME neurons (*Punc-25::mCherry*) and in GLR cells (*Pgly-18::mCherry*). *yadEx644* had a similar rescue ability as other transgenes expressing *unc-1*(cDNA) under the *Pdpy-30* promoter (*Figure 2B*). After screening 1037 transgenic animals, we were able to uncover 19 of them lost *unc-1 (+)* in RME neurons, 13 of them lacking *unc-1(+)* expression in GLR cells, and one animal without *unc-1(+)* expression in both RME neurons and GLR cells. None of these mosaic animals showed rescue of *unc-1(lf)* RME phenotypes (*Figure 2B*). These results suggest that gap junctions between GLR cells and RME neurons play critical roles in regulating RME axon specification.

One gap junction channel is composed of two hemichannels connected across the intercellular space between two cells. Eliminating the expression of gap junction channels in one of the two cells is likely to be sufficient to block gap junction formation. The innexin family of gap junction proteins has 25 members in *C. elegans*. Only two of them, *unc-7* and *unc-9,* are expressed in RME neurons, and ten innexins, including *unc-7,* appear to be expressed in GLR cells (*Altun et al., 2009*). To further confirm the importance of GLR-RME gap junctions in RME axon specification, we examined mutants of *unc-7* and *unc-9*. Single mutants of *unc-7* and *unc-9* had lower percentages of animals with SNB-1::GFP mislocalization defects when compared with *unc-1(lf)* animals, and *unc-7(lf)* phenotype was stronger than that of *unc-9(lf)* (*Figure 2C and D*). We further generated double mutants of *unc-7* and *unc-9* and found that these animals had the same degree of SNB-1::GFP defects as those in *unc-1(lf)* animals (*Figure 2C and D* and *Figure 1—figure supplement 2B*). Since *unc-7* is expressed in both GLR cells and RME neurons, it is likely that the *unc-7(lf)* phenotype is due to its roles in both cells. Indeed, expression of *unc-7* cDNA in GLR cells or RME neurons only partially rescued *unc-7(lf)* phenotypes, and expression of *unc-7* cDNA in both GLR cells and RME neurons completely restored SNB-1::GFP distribution in *unc-7(lf)* animals (*Figure 2E*). In addition to its function as a gap junction subunit, *unc-7* has been reported to regulate presynaptic development in a gap-junction-independent manner (*Yeh et al., 2009*). To exclude the possibility that the defects in

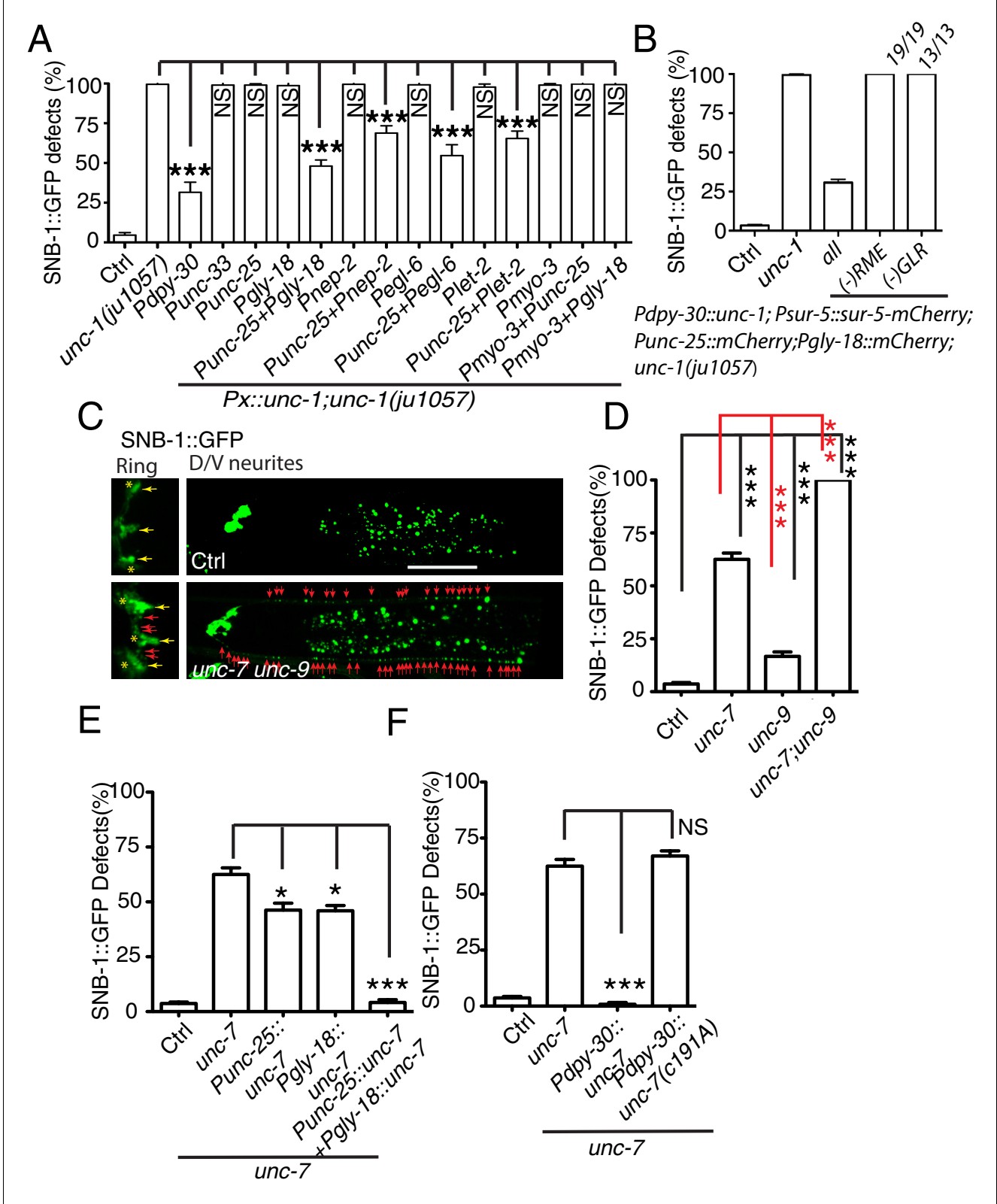

**Figure 2.** Gap junctions between GLR cells and RME neurons are required for RME axon specification. (**A**) Quantification of SNB-1::GFP puncta distribution defects in *unc-1(lf)* mutants and rescue transgenic strains. Promoters used to drive expression of *unc-1* cDNA are: *Pdpy-30* in all tissues; *Punc-33* and *Punc-25* in all neurons and RME neurons, respectively; *Pgly-18, Pnep-2, Pegl-6,* and *Plet-2* promoters in GLR cells (note: *Pgly-18, Pegl-6,* and *Plet-2* are also active in a few other cells that are not overlapping in identity) and *Pmyo-3* in muscle. (**B**) Results from mosaic analyses show that *unc-*
*Figure 2 continued on next page*

*Figure 2 continued*

1 is required in both GLR cells and RME neurons. (**C**) Representative images of SNB-1::GFP puncta distribution in control and *unc-7 unc-9* double mutant animals. (**D**) Quantification of SNB-1::GFP distribution defects in *unc-7*, *unc-9* single and *unc-7 unc-9* double mutant animals. (**E**) Quantification of SNB-1::GFP distribution defects in *unc-7* and rescued transgenic strains. (**F**) Quantification of SNB-1::GFP distribution defects in transgenic animals expressing wild type or C191A form of *unc-7*. Each experiment was performed with N > 200 animals at least three times. For transgenic animals, the results shown here are generated from at least three independent lines. Data are shown as mean ± SD. Student t-test, \*\*\*p<0.001, \*p<0.05, NS: no significant difference. Scale bar: 20 μm.

The following figure supplement is available for figure 2:

**Figure supplement 1.** GLR cells and RME neurons

*unc-7(lf)* animals were due to its non-gap-junction functions, we tested the rescue ability of a C191A mutant form of UNC-7. The C191A mutation has been shown to specifically interrupt UNC-7 gap junction functions, but not its ability to regulate presynaptic development (*Bouhours et al., 2011*). As shown in *Figure 2F*, ubiquitous expression of *unc-7*(C191A) (*Pdpy-30:: unc-7*(C191A)) failed to rescue *unc-7(lf)* phenotypes, while expression of wild-type *unc-7* cDNA (*Pdpy-30:: unc-7*) completely restored SNB-1::GFP distribution in *unc-7(lf)* animals. In conclusion, these data suggest that GLR-RME gap junctions play an important role in regulating RME axon specification.

## GLR-RME gap junctions regulate microtubule polarity through the CDK-5 pathway

Assembly and polarity of microtubule cytoskeleton display different properties in axons and dendrites. Axonal microtubules are all orientated with plus-end-out, while dendritic microtubules are either a mixture of plus-end-out and minus-end-out or minus-end-out only (*Barnes and Polleux, 2009*). The observation of misaccumulation of axonal markers in the gap junction mutants raises the question of whether GLR-RME gap junctions can regulate microtubule polarity in RME D/V neurites. Microtubule polarity can be visualized by imaging the dynamic of microtubule plus-end binding protein EBP-2::GFP, and the growing direction of EBP-2::GFP signal implicates the direction of microtubule plus-end (*Kozlowski et al., 2007*). Using an EBP-2::GFP reporter expressed only in RME neurons, we found that in RME D/V neurites of control animals, microtubules showed a mixture of plus-end-out (76%) and minus-end-out (24%) (*Figure 3A and B*). However, in *unc-1* and *unc-7 unc-9* mutants, almost all microtubules were plus-end-out (*unc-1:* 99%, *unc-7 unc-9:* 96%), which mirrors the microtubule assembly found in axons (*Figure 3A and B*). While GLR-RME gap junctions play important roles in regulating microtubule polarity, *unc-1(lf)* mutants do not seem to affect the distribution or the number of microtubules in RME D/V neurites. To visualize polymerized microtubules in RME neurons, we used EMTB::GFP (MT-binding domain of ensconsin fused to GFP), which binds along the side of polymerized microtubules and has been previously used to label microtubules in *C. elegans* neurons (*Richardson et al., 2014*). As shown in *Figure 3—figure supplement 1A and B*, *unc-1(lf)* did not change the overall distribution or intensity of EMTB::GFP in RME D/V neurites.

In *C. elegans* neurons, the CDK-5 pathway, consisted of *cdka-1/p35* and *cdk-5,* has been shown to regulate microtubule polarity (*Goodwin et al., 2012*; *Ou et al., 2010*). We found that *cdk-5(lf)* animals had similar microtubule polarity defects in RME D/V neurites as those in *unc-1(lf)* animals (*Figure 4A*), and loss-of-function mutations of *cdk-5* or its upstream regulator *cdka-1/*p35 induced misaccumulation of SNB-1::GFP in RME D/V neurites (*Figure 4B and C*). Expression of *cdk-5* cDNA in RME neurons rescued *cdk-5(lf)* phenotypes, supporting the conclusion that the CDK-5 pathway cell-autonomously regulates RME axon specification (*Figure 4C*). The similar phenotypes of gap junction and *cdk-5* mutants led us to test their genetic interactions. Similar to *unc-1(lf)* animals, almost all *unc-1;cdka-1* and *unc-1;cdk-5* animals had defects in RME axon specification (*Figure 3—figure supplement 1C*), and these double mutants showed a similar number of SNB-1::GFP puncta in RME D neurites as those in single mutants (*Figure 1—figure supplement 2B*), suggesting that the CDK-5 pathway genetically acts in the same pathway with GLR-RME gap junctions. We also analyzed the genetic interactions between *unc-7* and the CDK-5 pathway and found that double mutants of *unc-7;cdk-5* or *unc-7;cdka-1* had a similar degree of SNB-1::GFP mislocalization defects as those in *unc-7(lf)* single mutants, further supporting the conclusion that the CDK-5 pathway and GLR-RME

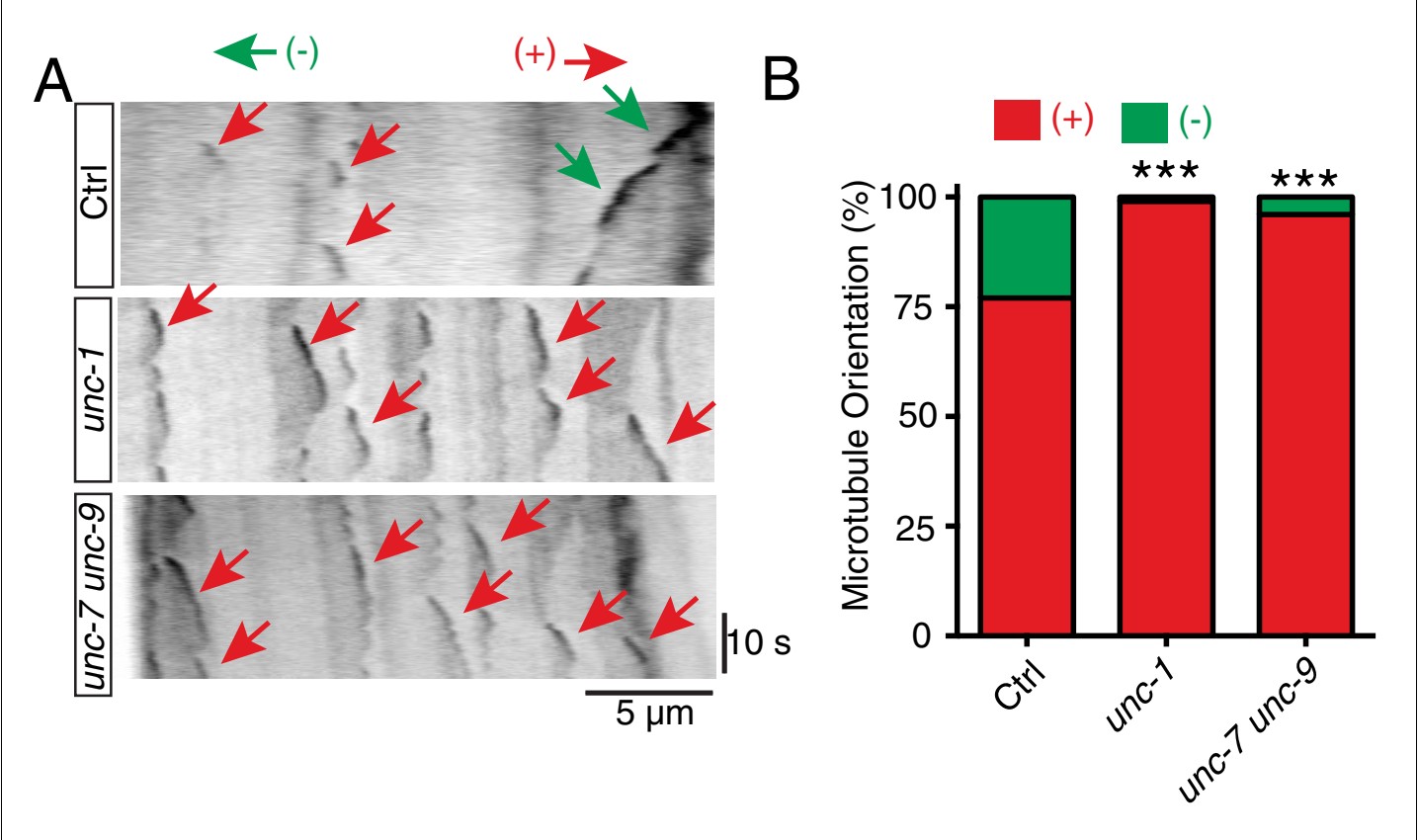

**Figure 3.** *unc-1* regulates microtubule polarity in RME D/V dendrites. (**A**) Time kymographs generated from a 40-s movie of EBP-2::GFP dynamics in control, *unc-1(e580)* and *unc-7 unc-9* animals. (**B**) Quantification of percentage of orientation in RME D/V neurites in control (n = 123), *unc-1(e580)* (n = 115) and *unc-7unc-9* animals (n = 110) based on the analysis of EBP-2::GFP movement. Values that differ significantly from wild type (Fisher's exact test) are denoted on the graphs (*p<0.05).

The following figure supplement is available for figure 3:

**Figure supplement 1.** *unc-1(lf)* does not change the amount of polymerized microtubules in RME D/V dendrites.

gap junctions function in the same genetic pathway (*Figure 4D*, *Figure 1—figure supplement 2B*). The activation of CDK-5 is regulated by p25, a cleavage product of p35, and the production of p25 has been used to measure the activation of the p35-CDK-5 pathway (*Dhavan and Tsai, 2001*). To test whether GLR-RME gap junctions are involved in activating the CDK-5 pathway in *C. elegans*, we generated a strain expressing FLAG-tagged p35 homolog CDKA-1 in RME neurons (*yadIs13*). In Western blot analyses, we detected a cleaved protein fragment of CDKA-1 in control animals (*Figure 4E*). Remarkably, the cleavage of CDKA-1 was largely eliminated in *unc-1* single and *unc-9 unc-7* double mutant animals, indicating that GLR-RME gap junctions indeed play an important role in activating the CDK-5 pathway in RME neurons (*Figure 4E and F*). Moreover, expression of mouse p25, the cleaved active version of p35, but not p35 in RME neurons suppressed *unc-1(lf)* phenotypes in a dose-dependent manner, further supporting that the *cdka-1*/p35-*cdk-5* pathway acts downstream of GLR-RME gap junctions (*Figure 4G*).

## GLR cells regulate the CDK-5 pathway through gap junction-mediated calcium influx

The cleavage of p35 to p25 is mediated by the calcium-dependent cysteine protease calpain (*Dhavan and Tsai, 2001*). To address the functional importance of calcium and calpain in the cleavage of CDKA-1/p35 in RME neurons, we treated *yadIs13* animals with the calcium antagonist

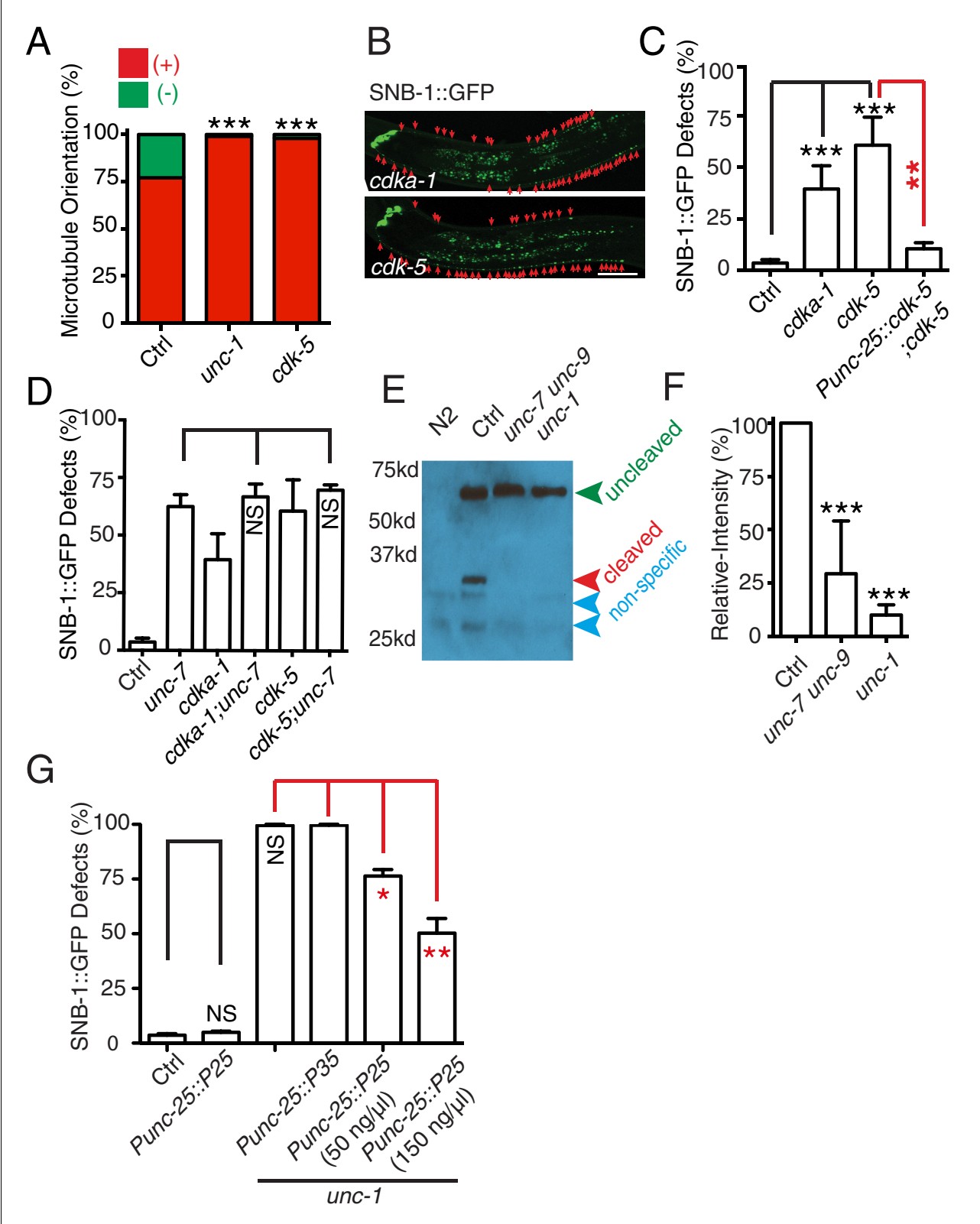

**Figure 4.** The CDK-5 pathway is regulated by GLR-RME gap junctions in RME axon specification (**A**) Loss-of-function of *cdk-5* induces microtubule polarity defects in RME D/V neurites to a similar degree as those in *unc-1(lf)* animals. *cdk-5*: n = 81. Values that differ significantly from wild type (Fisher's exact test) are denoted on the graphs (*p<0.05). (**B**) Representative images and (**C**) quantification of SNB-1::GFP distribution defects in *cdka-1* (*lf*), *cdk-5(lf)*, and *cdk-5(lf)* transgenic rescued strains. (**D**) Quantification of SNB-1::GFP distribution defects in *unc-7;cdka-1* and *unc-7;cdk-5* double

Figure 4 continued

mutants. (E) Western blot results of FLAG::CDKA-1 (*yadIs13: Punc-25::FLAG::cdka-1(cDNA)*) expression in non-transgene N2, transgene control, *unc-7 unc-9* and *unc-1(e580)* animals. Red arrowhead points to the cleaved band. Green arrowhead marks to the uncleaved band. Blue arrowheads highlight nonspecific bands recognized by the anti-FLAG antibody. (F) Quantification of effect of *unc-7unc-9* and *unc-1(e580)* on CDKA-1 cleavage. Three independent experiments were performed. Band intensity was measured using imageJ, and the relative intensity was calculated by $(I_{cleaved} / I_{uncleaved})_{mutant} / (I_{cleaved} / I_{uncleaved})_{control} \times 100\%$. $I_{cleaved}$ or $I_{uncleaved}$ = intensity of cleaved CDKA-1/ intensity of un-cleaved CDKA-1 in mutant or control animals. (G) Quantification of SNB-1::GFP distribution defects in transgenes expressing p25 or p35 in *unc-1(ju1057)* background. In C, D and G, experiments were performed with N > 200 animals at least three times. For transgenic animals, the results shown here are generated from at least three independent lines. Data are shown as mean ± SD. Student t-test, *p<0.05, **p<0.01, ***p<0.001. Scale bar: 20 μm.

BAPTA-AM and calpain-specific inhibitor PD150606 as previously described (*Ghosh-Roy et al., 2010*; *Xu and Chisholm, 2011*). The cleavage of CDKA-1/p35 was suppressed by both BAPTA-AM and PD150606 treatments (*Figure 5A and B*), which confirms the important role of calcium and calpain in the regulation of the CDK-5 pathway in RME neurons. In animals treated with BAPTA-AM or PD150606, we observed a similar SNB-1::GFP distribution defect as seen in *cdka-1(lf)* and *cdk-5(lf)* animals (*Figure 5C*). Treating animals with BAPTA-AM decreases calcium level in many cells and affects the overall condition of animals, so it could be argued that the effect of BAPTA-AM treatment may be caused by the sickness of animals. Previous studies showed that expression of a calcium buffer protein calbindin D28K in *C. elegans* neurons could lower intracellular calcium concentration (*Schumacher et al., 2012*). To show the importance of calcium in RME neurons, we expressed calbindin D28K in RME neurons and found that such expression caused RME defects at a similar level as BAPTA-AM treated animals (*Figure 5D*). More importantly, expression of calbindin D28K only in GLR cells (*Pgly-18* and *Pegl-6*) caused the same extent of defects as animals expressing calbindin D28K in RME neurons or in both GLR cells and RME neurons (*Figure 5D*). As a control, expression of calbindin D28K in muscle did not cause any obvious RME defects (*Figure 5D*). These results support that the GLR cells can regulate RME calcium concentration through GLR-RME gap junctions. To further illustrate the effect of lowering GLR calcium on RME axon specification, we tested microtubule polarity in animals expressing calbindin D28K in GLR cells. Consistent with our observation using the SNB-1::GFP marker, microtubule polarity was changed to be mostly plus-end-out in D/V neurites in these transgenic animals (*Figure 5E* plus-end-out: 96%). Furthermore, mutating *cdk-5* in transgenicanimals expressing calbindin D28K in GLR cells did not enhance the RME SNB-1::GFP phenotypes, and activating the CDK-5 pathway in RME neurons by overexpressing p25 suppressed the phenotypes in those transgenic animals (*Figure 5F*). Taken together, we show that GLR cells can regulate RME calcium concentration through gap junctions, and such regulation can modulate the activation of the CDK-5 pathway to affect RME axon specification.

To reveal which calpain is involved in RME axon specification, we tested all seven calpains (*clp-1-clp-7*). We found that *clp-4(lf)* displayed a similar degree of RME defects as seen in *cdk-5(lf)* mutants, while mutations or RNAi of others did not produce obvious phenotypes (*Figure 5D*). Our results also showed that *clp-4* cell-autonomously regulates RME axon specification (*Figure 5D*). Double mutants of *clp-4* with *unc-1* or *unc-7* displayed RME polarity defects to a similar extent as those in single mutants, supporting the conclusion that *clp-4* functions in the GLR-RME gap junction-*cdk-5* pathway to regulate RME axon specification (*Figure 5E*, *Figure 1—figure supplement 2B*).

With the finding of the important role of calcium coming from GLR cells, we further addressed whether it played a permissive or instructive role in RME axon specification. *slo-1* encodes a voltage-gated potassium channel, and mutation of *slo-1* has been shown to increase neuronal activity, possibly calcium (*Wang et al., 2001*). We found that *slo-1(lf)* partially suppressed RME phenotypes in *unc-1(lf)* and transgenic animals expressing calbindin D28K in GLR cells (*Figure 6A and B*). Moreover, *slo-1(lf)* suppressed both the microtubule polarity defects and the blockage of CDKA-1 cleavage in *unc-1(lf)* animals (*Figure 6B,C and D*). These results suggest that calcium influx from GLR cells likely plays a permissive rather than an instructive role in RME axon specification.

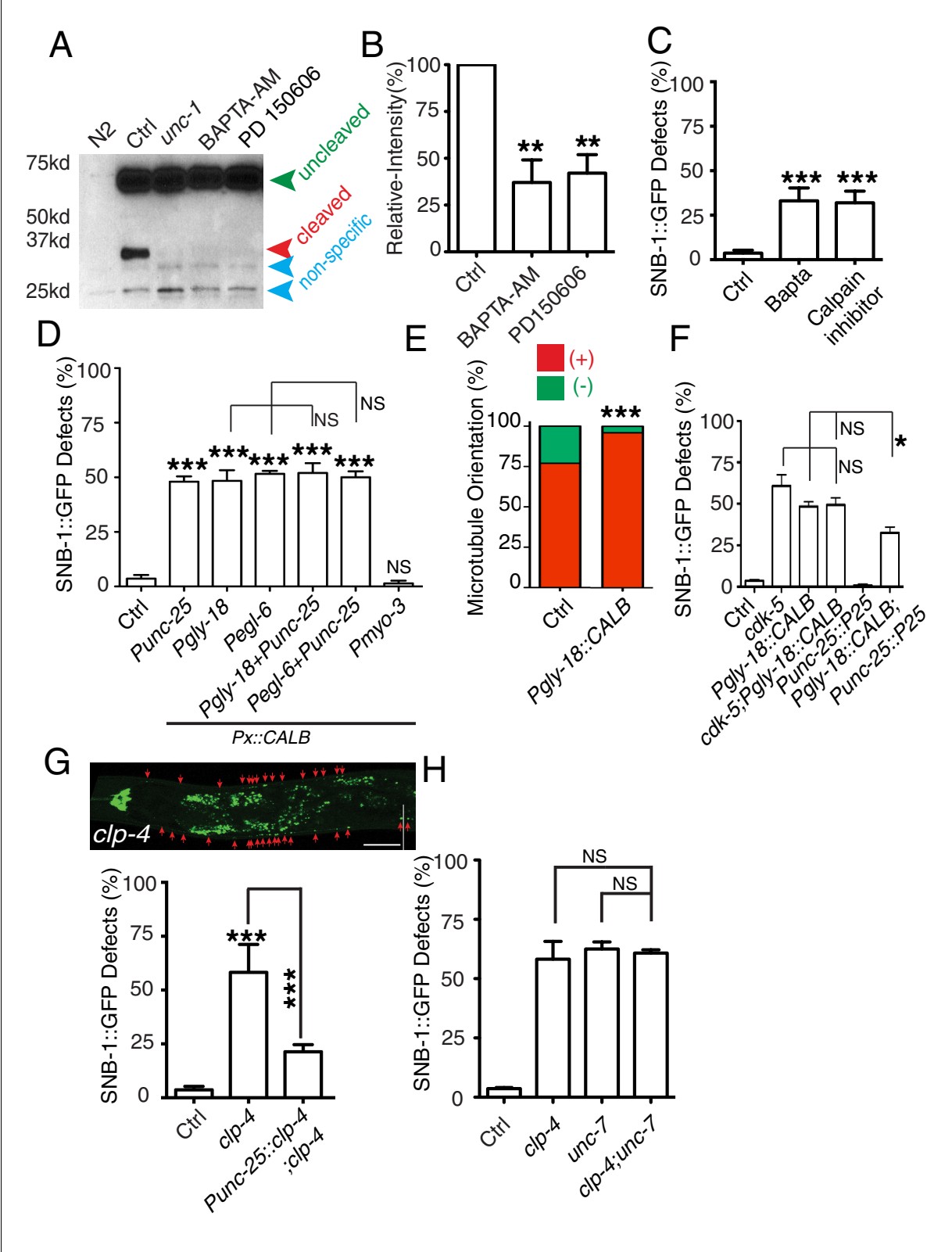

**Figure 5.** GLR-RME gap junctions regulate the CDK-5 pathway through calcium and calpain. (**A**) Gel images and (**B**) quantification results show that calcium antagonist BAPTA-AM and Calpain inhibitor PD150606 suppress the cleavage of CDKA-1/p35. Three independent experiments were performed, and the relative intensity was calculated as described in *Figure 4* (**C**) Quantification of SNB-1::GFP distribution defects in BAPTA-AM and Calpain inhibitor PD150606-treated animals. (**D**) Quantification of SNB-1::GFP distribution defects in animals expressing calbindin D28K (CALB) in RME

*Figure 5 continued on next page*

*Figure 5 continued*

neurons (*Punc-25*), GLR cells (*Pgly-18* and *Pegl-6*), and muscle (*Pmyo-3*). For this experiment and all other experiments using calbindin D28K transgenes, only animals with normal GLR cell and RME neuron morphology were chosen for quantification. (E) Expression of calbindin D28K (CALB) in GLR cells (*Pgly-18*) induces microtubule polarity defects in RME D/V neurites. n = 100. Values that differ significantly from wild type (Fisher's exact test) are denoted on the graphs ***p<0.001. (F) Quantification of SNB-1::GFP distribution defects in animals expressing calbindin D28K (CALB) in GLR cells (*Pgly-18*) in wild-type background, in *cdk-5(lf)* background, or in a transgene backgound overexpressing p25 in RME neurons (*Punc-25*). (G) Quantification of SNB-1::GFP distribution defects in *clp-4* mutants and rescue transgenes . (H) Quantification of SNB-1::GFP distribution defects in *clp-4*, *unc-7*, and *clp-4;unc-7* double mutants. In C, D, and E, data are shown as mean ± SD. Student test, *p<0.05, **p<0.01, ***p<0.001. NS: no significant difference.

## Discussion

Our data show that in *C. elegans*, GLR cells can regulate axon specification of nearby RME neurons through gap junctions. Specifically, GLR-RME gap junctions can modulate the strength of the CDK-5 pathway to regulate microtubule polarity in RME D/V neurites. Two lines of evidence show that RME axon specification is interrupted in gap junction mutants: first, using well-established presynaptic vesicle and active zone markers, we found that these axonal components were mis-accumulated in non-axonal neurites; second, by imaging microtubule dynamics, we revealed that microtubule polarity in RME D/V neurites was changed from a mixture of plus-end-out and minus-end-out orientations to a predominantly axon-like plus-end-out orientation. Our analyses of the cleavage of CDKA-1/p35 show that GLR-RME gap junctions can regulate the activation of the CDK-5 pathway in a calcium-dependent manner. From these data, we propose that during development, the influx of calcium from GLR cells to RME neurons through gap junctions can induce the activation of the CDK-5 pathway to guide microtubule assembly and axon specification (*Figure 7*).

### Synapse elimination and axon specification

In our previous studies, we show that RME neurons form transient synapses at the end of D/V neurites at the L1 stage, and the cell death (CED) pathway can regulate the elimination of those transient synapses through cleavage of a F-actin severing protein GSNL-1/gelsolin (*Meng et al., 2015*). CED mutants (*egl-1(lf)*, *ced-9(gf)*, *ced-3(lf)* and *ced-4(lf)*) or *gsnl-1(lf)* prevent the elimination of transient synapses in D/V neurites, and as a result SNB-1::GFP puncta are observed at the end of D/V neurites in adult animals. The phenotypes of gap junction mutants (*unc-1*, *unc-7*, *unc-9*, *unc-7 unc-9*) are different from those of CED and *gsnl-1* mutants. Loss of function in CED or *gsnl-1* animals do not show any difference with control animals at the L1 stage in that they have similar numbers of SNB-1::GFP puncta at the end of D/V neurites. In contrast, gap junction mutant animals reported here already exhibit differences with control animals in the L1 stage, as they have SNB-1::GFP puncta along the D/V neurites in addition to those at the end of D/V neurites. When animals reach the adult stage, CED or *gsnl-1(lf)* animals only have SNB-1::GFP at the end of D/V neurites, but gap junction mutants have SNB-1::GFP along the entire D/V neurites. While gap junction mutants have defects of SNB-1::GFP distribution in axonal processes, no such phenotypes were observed in CED or *gsnl-1* mutants. Different from gap junction mutants, CED or *gsnl-1* mutant animals have similar microtubule polarity in D/V neurites as that in control animals (data no shown). With these differences and other evidence presented in this study, we conclude that GLR-RME gap junctions are involved in regulating RME axon specification rather than synapse elimination.

### Functions of neuron-glia gap junctions

A series of ultrastructure and light microscopy studies have demonstrated that there are gap junction structures between neurons and glial cells (*Cuadras et al., 1985*; *Nadarajah et al., 1997*; *Nadarajah and Parnavelas, 1999*; *Nadarajah et al., 1996*; *Peinado et al., 1993*; *Shivers et al., 1988*; *Sinues Porta et al., 1988*). Several studies using electrophysiological and dye-filling methods support that those structures are likely functional gap junctions (*Alvarez-Maubecin et al., 2000*; *Froes et al., 1999*; *Nedergaard, 1994*; *Nedergaard and Goldman, 1993*). In *C. elegans*, the presence of gap junctions between RME neurons and GLR cells has been well documented (*White et al., 1986*). GLR cells are classified as glial cells in *C. elegans* (*Oikonomou and Shaham, 2011*), but since GLR cells are derived from a muscle rather than neural lineage, some studies classify them as a part

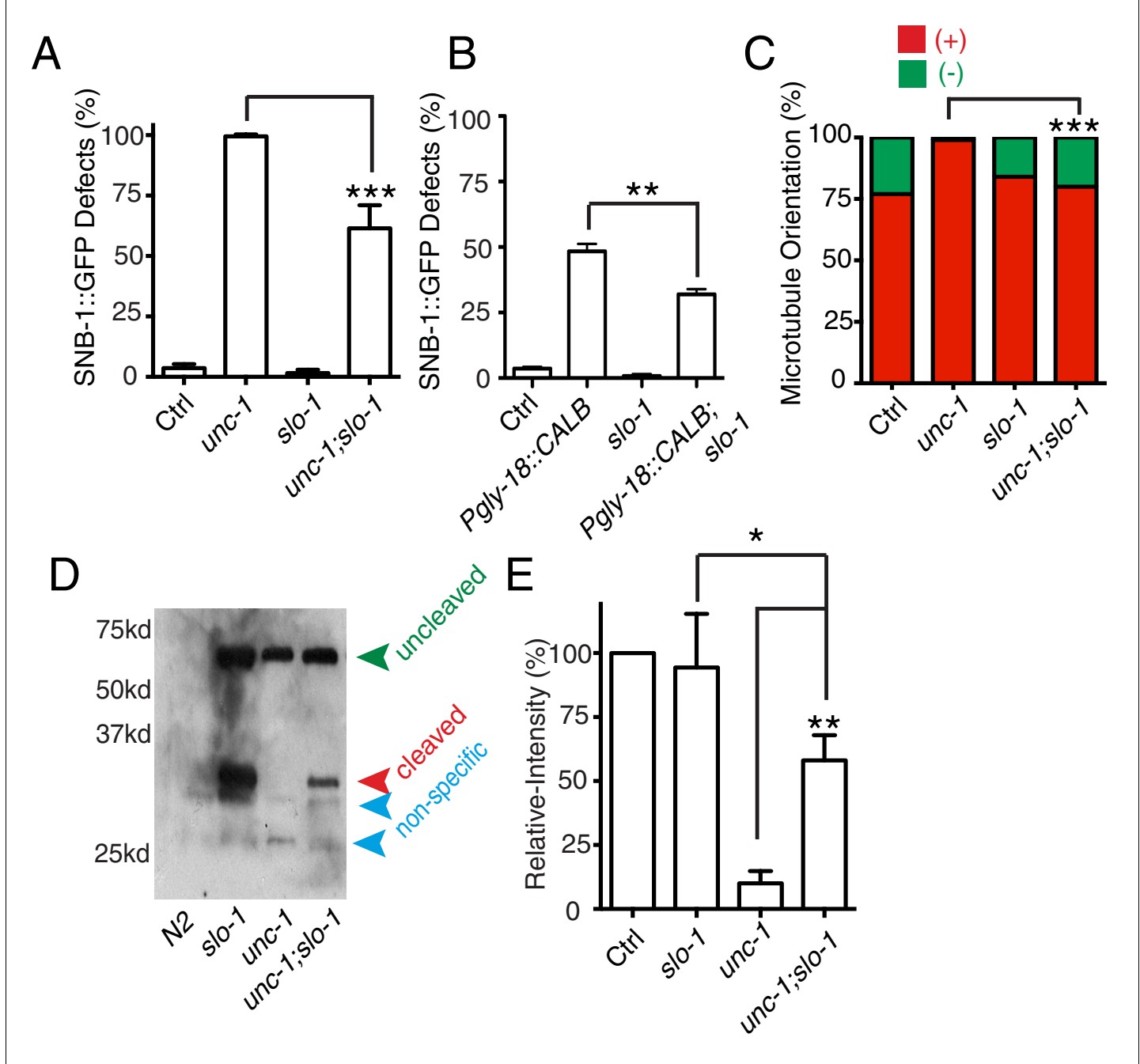

**Figure 6.** *slo-1(lf)* suppresses *unc-1(lf)* phenotypes through the CDK-5 pathway. (**A**) Quantification of SNB-1::GFP distribution defects in *unc-1(ju1057)*, *slo-1* and *unc-1(ju1057);slo-1.* (**B**) Quantification of SNB-1::GFP distribution defects in animals expressing calbindin D28K (CALB) in GLR cells (*Pgly-18*) in a control or *slo-1(lf)* background. (**C**) Quantification of EBP-2 dynamics show that loss-of-function of *slo-1* suppresses microtubule polarity defects in RME D/V neurites in *unc-1(e580)* animals. Values that differ significantly from wild type (Fisher's exact test) are denoted on the graphs (*p<0.05). (**D**) Western blot results of FLAG::CDKA-1 (*yadls13*) expression in non-transgene N2, *slo-1*, *unc-1*, and *unc-1;slo-1* animals. Red arrowhead points to the cleaved band. Green arrowhead marks the uncleaved band. Blue arrowheads highlight nonspecific bands recognized by the anti-FLAG antibody. (**E**) Quantification of western results shows that loss-of-function of *slo-1* suppresses the *unc-1* effects on CDKA-1 cleavage. Three independent experiments were performed, and quantification uses the same method as in *Figure 4* In **A** and **B**, experiments were performed with N > 200 animals at least three times. For transgenic animals, the results shown here are generated from at least three independent lines. Data are shown as mean ± SD. Student t-test, *p<0.05, **p<0.01, ***p<0.001.

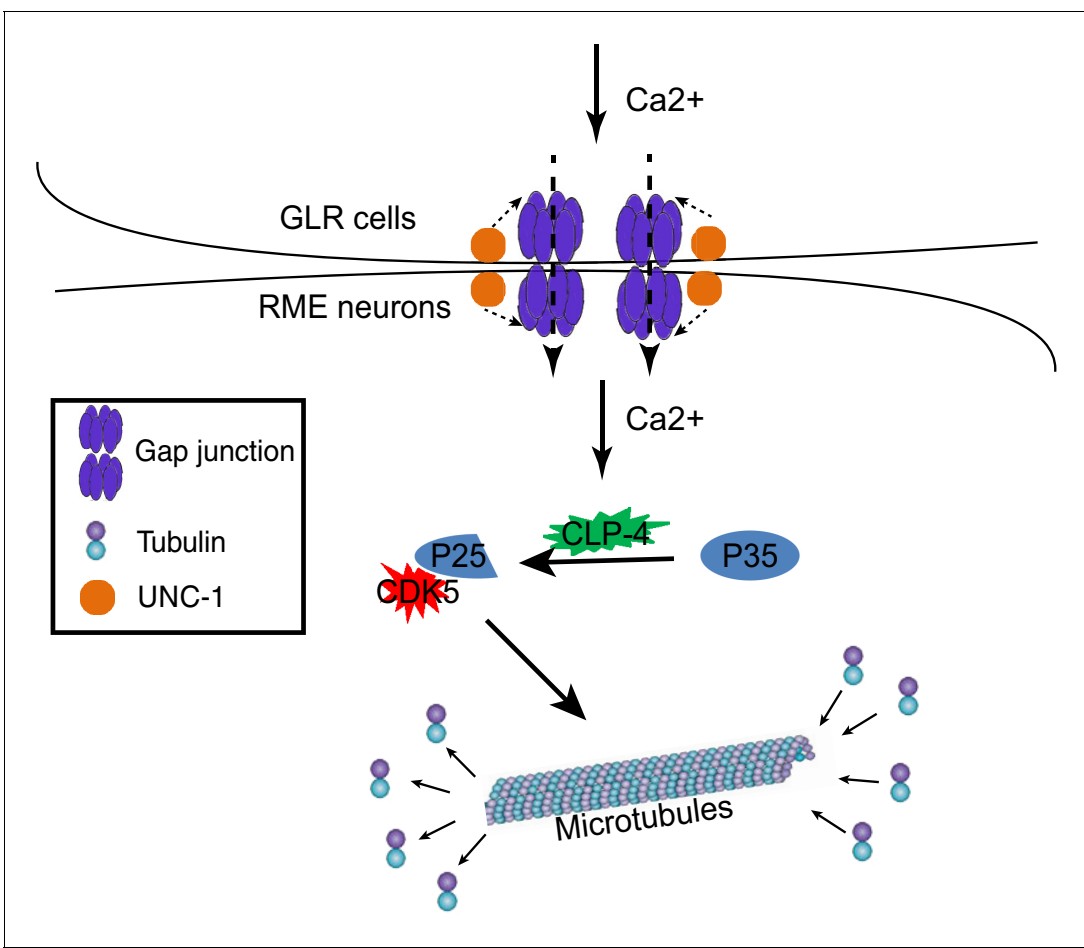

**Figure 7.** A model for GLR-RME gap junction regulated RME axon specification. At the early developmental stage, calcium influxes from GLR cells to RME neurons through gap junctions to induce activation of the CDK-5 pathway, which guides microtubule assembly and axon specification.

of the mesodermal/muscle system instead. In the mammalian cortex, most neurons undergo axon-dendrite polarization during migration along the radial glia (*Barnes and Polleux, 2009*). Radial glia, neural precursors, and immature neurons all express multiple connexins, and gap junctions between radial glia and between neurons have been shown to be critical in neurogenesis and circuit assembly (*Belousov and Fontes, 2013*; *Connors and Long, 2004*; *Kunze et al., 2009*; *Li et al., 2012*; *Yu et al., 2012*). Radial glia are also coupled with neural precursors through gap junctions at the ventricular and subventricular zones (*Bittman et al., 1997*), raising the possibility that radial glia may regulate neural precursors through gap junctions.

## Signals affecting axon specification/neuronal polarity

Using the classic in vitro culture system developed by Banker *et al.* (*Craig and Banker, 1994*; *Goslin and Banker, 1989*), numerous studies have revealed that asymmetric activation of intracellular signaling pathways is the major driving force in determining the fate of axons and dendrites (*Barnes and Polleux, 2009*). Recent evidence also shows that extracellular cues play instructive roles in regulating neuronal polarity. In cultured hippocampal neurons plated on striped substrates coated with Laminin and NgCAM, the first neurite that contacts the boundary between two stripes with different substrates always becomes the axon regardless of the initial substrate that the neurite grows on (*Esch et al., 1999*). Extracellular BDNF could determine which neurites will become the axon by activating the LKB1-SAD pathway in cultured neurons and the mammalian cortex (*Barnes et al., 2007*; *Shelly et al., 2007*). In *Xenopus* retinal ganglion cells, extracellular cues from basal lamina play an essential role in establishing neuronal polarity (*Zolessi et al., 2006*). Furthermore, studies in

*C. elegans* highlight the important roles of guidance cues, such as Wnt and Netrin, in instructing neuronal polarity (*Hilliard and Bargmann, 2006*; *Poon et al., 2008*; *Prasad and Clark, 2006*). Through genetic analyses of RME development in *C. elegans*, we reveal that the gap junctions between GLR cells and their nearby RME neurons play an important role in regulating RME axon specification by activating the CDK-5 pathway.

## Gap junction signals in regulating neuronal polarity

Gap junction-mediated calcium signaling plays important roles in development and disease (*Chuang et al., 2007*; *Nakase and Naus, 2004*; *Schumacher et al., 2012*). Our results highlight the critical role of glia-neuron gap junctions in regulating a calcium-dependent CDK-5 pathway in axon specification. Besides calcium, gap junctions can mediate the exchange of small molecules that are less than 1 kDa, such as IP3, cAMP, and cGMP (*Goodenough and Paul, 2009*). Gap junction channels also directly bind to a large group of proteins, including Zona occludens-1(ZO-1), SRC, PKA, PKC, MAP Kinases, Cadherin, GTPase, phosphatases, microtubules, and F-actin (*Giepmans, 2004*). The relatively weaker phenotypes observed in mutants affecting the *clp-4*/calpain-p35-CDK-5 pathway suggest that GLR-RME gap junctions may regulate multiple downstream pathways. Given that many regulators of neuronal polarity, such as cAMP, cGMP, Cadherin, MAP kinases, and GTPases, have been shown to be modulated by gap junctions (*Barnes and Polleux, 2009*; *Giepmans, 2004*; *Goodenough and Paul, 2009*), it is possible that GLR-RME gap junctions may regulate some of those pathways as well.

# Materials and methods

## *C. elegans* genetics

We maintained *C. elegans* strains on NGM plates at 20–22.5°C. All transgenes and strains are described in *Supplementary file 1*. We used *juIs1* (*Punc-25::SNB-1::GFP*) in *unc-30* (*ju32*) background to visualize RME neuron presynaptic terminals (*Meng et al., 2015*). *hpIs3 (Punc-25::SYD-2::GFP*) (*Yeh et al., 2009*) was used to visualize RME presynaptic active zones. *unc-1(ju1057)* was isolated from the strain CZ15720 *unc-30(ju32) (IV) juIs1(Punc-25::SNB-1::GFP) (IV)* after ethyl methane sulfonate mutagenesis. In experiments examining the localization of SYD-2::GFP, RAB-3::mCherry and EBP-2::GFP dynamics, we used another *unc-30* allele *ok613*.

## Drug experiments

BAPTA-AM and PD150606 were obtained from Sigma and were dissolved in DMSO before dilution in M9. We treated animals with drugs as previously described (*Ghosh-Roy et al., 2010*; *Xu and Chisholm, 2011*). For Western blot, we incubated mixed-stage animals in the drug solutions (containing Escherichia coli OP50) for 24 hr before experiments; for quantification of RME defects, we grew L1 animals in the drug solutions (containing *Escherichia coli* OP50) until they reached the adult stage (36–48 hr). Control animals were incubated in M9-containing DMSO solvent to the equivalent concentration as that used in the specific experiment.

## Temperature shift experiments

For quantification of *unc-1(hs1)* phenotypes at 11°C and 25°C, animals were grown at 11°C/25°C for at least two generation before examining their phenotypes. 11°C->(L1) 25°C: L4 animals were cultured in 11°C, and in the next generation, adult animals were collected for synchronized eggs. Eggs were hatched at 11°C and then transferred to 25°C until examination of the phenotype at the day 1 adult stage. 25°C->(L1) 11°C->(L4) 25°C, animals were grown at 25°C for two generations, and then L1 animals were transferred to 11°C. When animals reached middle L4 stages at 11°C, those animals were transferred back to 25°C until examination of the phenotype at the day 1 adult stage. 25°C-> (L4) 11°C: similar as described in the '11°C->(L1) 25°C' section, but using 25°C as the starting temperature and 11°C as the final temperature, and animals were transferred at L4 stage.

## Cloning and constructs

All DNA expression constructs were made using Gateway cloning technology (Invitrogen, Carlsbad, CA). Sequences of the final clones were confirmed. *Supplementary file 1*. lists the genotypes. *unc-1*

and *unc-9* cDNA were amplified from yk364C8 and yk1324e06. *unc-7*, *ebp-2*, *cdka-1,* and *clp-4* cDNA were amplified from a homemade N2 cDNA pool. DNA construct information is listed in *Supplementary file 2.* Transgenic animals were generated following standard procedures. In general, plasmid DNAs of interest were used at 1–50 ng/µl with the co-injection marker *Pttx-3::RFP* at 40 ng/µl.

## Fluorescence microscopy

Representative images were acquired with a Zeiss LSM700 confocal microscope using a Plan-Apochromat 40x/1.4 objective. Worms were immobilized in 1% 1-phenoxy-2-propanol (TCI America, Potland, OR) in M9 buffer. For quantification of polarity defects in RME neurons, we used a Zeiss Axio Imager 2 microscope equipped with Chroma HQ filters. RME axon specification defects were scored for the percentage of animals with evenly distributed SNB-1::GFP puncta (more than 25) in both RME D/V dendrites. Each condition consisted of at least three independent experiments and 200–300 1-day-old adults. For quantification of the number of SNB-1::GFP puncta in RME D neurites, we used the Zeiss Axio Imager 2 microscope with 63X objective. We selected animals with the dorsal side up and manually counted the number of all visible SNB-1::GFP puncta. For analysis of SNB-1:: GFP intensity, as shown in *Figure 1—figure supplement 1A*, we measured the intensity of each region as labeled in the *Figure 1—figure supplement 1* and then subtracted by the background of the same size area close to regions of interest. The fractions of GFP signals were calculated by dividing each part with the sum of all GFP signals.

## Protein analysis

For the CDKA-1 cleavage study, we generated an integrated transgene (*yadIs13*) expressing FLAG-CDKA-1 under the control of Pan-neuronal promoter *Prgef-1*. Proteins from mixed-stage animals were first extracted using sample buffer by freezing and thawing about 20 times, which consists of freezing in ethanol with dry ice followed by incubation at 95°C for 5 min. Protein samples were separated using SDS-PAGE Gradient Gels (4–20%), and then transferred to nitrocellulose. Blots were probed with rabbit anti-Flag antibodies (Sigma, F1804), and visualized with Amerisham HRP-conjugated anti-rabbit secondary antibodies at 1:5000 using the SuperSignal West Femto kit (Pierce, Rockford, 1L).

## Dynamic imaging

EBP-2::GFP dynamic experiments were performed using an Andor revolution microscope with a 60 x/1.46 Plan-Apochromat objective controlled by the MetaMorph software. All videos were acquired by an Andor EM-CCD camera (DU897). Young adult animals were immobilized in 5 mM levamisole and on 5% agar pads for imaging. Videos for EBP-2::GFP dynamic analysis were roughly 30 s with 10 frames per second. Kymographs were generated using ImageJ.

## Statistical analysis

We analyzed our data using one-tailed Student's t-test, one-way ANOVA, and Fisher's exact test in Graphpad Prism (GraphPad Software, La Jolla, CA).

## Acknowledgements

Some strains used in this study were provided by the Caenorhabditis Genetics Center (CGC), which is funded by NIH Office of Research Infrastructure Programs (P40 OD010440). We thank Dr. Y Kohara for *unc-1* and *unc-9* cDNAs, Dr. Peter Juo for EBP-2::GFP plasmid, Dr. Chiou-Fen Chuang for calbindin D28K plasmid, and Dr. Mei Zhen for *unc-7 unc-9* double mutants. We thank Dr. Cagla Eroglu and our lab members for comments on the manuscript. Y J is an Investigator of the Howard Hughes Medical Institute. This project is supported by a NIH R01 grant (NS094171 to DY).

## Additional information

### Funding

| Funder | Grant reference number | Author |
|---|---|---|
| National Institute of Neurological Disorders and Stroke | NS094171(R01) | Dong Yan |
| Duke University School of Medicine | faculty startup | Dong Yan |
| National Institute of Neurological Disorders and Stroke | NS076646 (R00) | Dong Yan |

The funders had no role in study design, data collection and interpretation, or the decision to submit the work for publication.

### Author contributions

LM, DY, Conception and design, Acquisition of data, Analysis and interpretation of data, Drafting or revising the article; AZ, Acquisition of data, Analysis and interpretation of data, Drafting or revising the article; YJ, Provided inputs and edited the manuscript, Drafting or revising the article

### Author ORCIDs

Yishi Jin, http://orcid.org/0000-0002-9371-9860
Dong Yan, http://orcid.org/0000-0002-7542-1251

## Additional files

### Supplementary files

• Supplementay file 1. Supplemental Table 1, Strain list.

• Supplementay file 2. Supplemental Table 2, Plasmid list.

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
