## [Decision Letter]

Thank you for submitting your article "Regulation of Neuronal Axon Specification by Glia-Neuron Gap Junctions in *C. elegans*" for consideration by *eLife*. Your article has been favorably evaluated by Marianne Bronner (Senior Editor) and two reviewers, one of whom is a member of our Board of Reviewing Editors. The reviewers have opted to remain anonymous.

The reviewers have discussed the reviews with one another and the Reviewing Editor has drafted this decision to help you prepare a revised submission.

Specifically, the reviewers expressed enthusiasm about the problems addressed in this paper and find the results of potential interest. However, the reviewers agreed that there are two issues that need to be fixed in order to support the author's conclusions:

1) The authors have not provided any evidence of the key claim of the paper, namely that GLR cells provide calcium via gap junction to control RME axon specification. Their argument rests entirely on showing that a calcium-dependent pathway is required for axon specification, but whether GLR neurons provides calcium is not tested. One way to do this would be to titrate out calcium in GLR by expressing a calcium binding protein in the GLR neurons (see this paper for experiment to test calcium trafficking through gap junctions: http://dev.biologists.org/content/139/22/4191.short).

2) There are some issues with the *gly-18* and *nep-2* promoter specificity that the authors regrettably fail to mention: both promoters are also active in muscle cells (plus a good number of additional cells) and hence the rescue experiments performed do not proof GLR involvement. This is disappointing because there appear to be a number of more suitable drivers in the literature (for example, *egl-6*) that the authors should have used. The authors need to use drivers with greater specificity.

On a purely editorial level, the reviewers would also appreciate if the authors could integrate – in the Discussion section – their present findings with their previous finding which demonstrated (a) that RME neurites initially do accumulate presynaptic clusters; (b) the existence of a CED-3-based mechanism to remove these clusters. What are the potential link between these mechanisms? Along these lines, it appears confusing that the author use the word "axon specification" in this paper, given that the process that the study is really a pruning process. Are the authors thinking that there is no polarity initially, the CDK pathway establishes this polarity and CED-3 then cleans up the "mess" that existed previously?

The authors will also need to have the manuscript undergo a substantial grammar check.

---

## [Author Response]

*[…] Specifically, the reviewers expressed enthusiasm about the problems addressed in this paper and find the results of potential interest. However, the reviewers agreed that there are two issues that need to be fixed in order to support the author's conclusions:*

*1) The authors have not provided any evidence of the key claim of the paper, namely that GLR cells provide calcium via gap junction to control RME axon specification. Their argument rests entirely on showing that a calcium-dependent pathway is required for axon specification, but whether GLR neurons provides calcium is not tested. One way to do this would be to titrate out calcium in GLR by expressing a calcium binding protein in the GLR neurons (see this paper for experiment to test calcium trafficking through gap junctions: http://dev.biologists.org/content/139/22/4191.short).*

Thanks for the suggestion. We generated transgenes expressing calbindin D28K in GLR cells using two different promoters (*Pgly-18* and *Pegl-6*) or RME neurons. As shown in Figure 5, expression of calbindin D28K in GLR cells caused defects in RME axon specification to a similar level as expression of calbindin D28K in RME neurons or in both RME neurons and GLR cells, supporting the conclusion that titrating out calcium in GLR cells could lower calcium level in RME, thereby affecting RME axon specification. We also showed that expression of calbindin D28K in GLR cells changed microtubule polarity in RME D/V neurites (Figure 5). As a control, expression of calbindin D28K in muscle did not cause discernable RME defects. To address whether calcium from GLR cells could regulate the CDK-5 pathway in RME neurons, we expressed calbindin D28K in GLR cells in *cdk-5(lf)* animals and observed a similar extent of defects as those in *cdk-5(lf)* or in GLR calbindin D28K transgene animals in wild type (Figure 5). These data are consistent with that the calcium from GLR cells likely functions in the same pathway with the CDK-5 pathway. Furthermore, we showed that activation of the CDK-5 pathway in RME neurons by overexpressing p25, the active form of p35, suppressed the phenotypes caused by overexpression of calbindin D28K in GLR cells (Figure 5). These lines of evidence support the conclusion that GLR cells can provide calcium through gap junctions to regulate the CDK-5 pathway in RME neurons and to affect RME axon specification.

In those experiments, we also noticed that expression of calbindin D28K in GLR cells caused lethality in ~10-20% of animals (multiple lines), and abnormal GLR morphology in ~25-40% of animals (multiple lines). In animals with abnormal GLR morphology, all showed defects in RME axon specification. To quantify the effects of calbindin D28K expression on RME axon specification, we only considered animals with normal GLR morphology, and we have described how we quantified the phenotypes in the Figure 5 legend. Similarly, in RME expressing calbindin D28K transgenes, ~40-50% of animals (multiple lines), have abnormal RME morphology. We only quantified animals with normal RME morphology (Figure 5 legend). We tried to generate integrated transgene animals expressing calbindin D28K in GLR cells to analyze CDKA-1 cleavage using Western blot, but all integrated transgene lines were very sick with strong lethality phenotypes. Those phenotypes are likely due to the essential function of GLR cells in worm development (http://www.wormatlas.org/hermaphrodite/muscleGLR/MusGLRframeset.html).

The difficulty in maintaining the integrated transgene animals precluded us to test the effect of titrating out calcium from GLR cells on CDKA-1 cleavage.

Nevertheless, our genetic analyses placed the CDK-5 pathway downstream of calcium coming from GLR cells.

*2) There are some issues with the gly-18 and nep-2 promoter specificity that the authors regrettably fail to mention: both promoters are also active in muscle cells (plus a good number of additional cells) and hence the rescue experiments performed do not proof GLR involvement. This is disappointing because there appear to be a number of more suitable drivers in the literature (for example, egl-6) that the authors should have used. The authors need to use drivers with greater specificity.*

In the revised manuscript, we added two other promoters *Pegl-6* and *Plet-2* to express *unc-1* in GLR cells, and expression of *unc-1* under these two promoters showed similar effects as those using *Pgly-18 and Pnep-2* (Figure 2). We did notice that *Pgly-18, Pegl-6* and *Plet-2* drove expression in other cells such as neurons and muscle, but our *Pnep-2::mCherry* transgene had noticeable expression only in GLR cells (Figure 2—figure supplement 1). Since GLR cells are the only common cells where those four promoters can drive expression of transgenes, we conclude that expression of *unc-1* in GLR cells is required for RME axon specification.

*On a purely editorial level, the reviewers would also appreciate if the authors could integrate – in the Discussion section – their present findings with their previous finding which demonstrated (a) that RME neurites initially do accumulate presynaptic clusters; (b) the existence of a CED-3-based mechanism to remove these clusters. What are the potential link between these mechanisms? Along these lines, it appears confusing that the author use the word "axon specification" in this paper, given that the process that the study is really a pruning process. Are the authors thinking that there is no polarity initially, the CDK pathway establishes this polarity and CED-3 then cleans up the "mess" that existed previously?*

We added a Discussion section in the revised manuscript. In our previous studies, we show that RME neurons form transient synapses at the end of D/V neurites at the L1 stage, and the cell death pathway can regulate the elimination of those transient synapses through cleavage of a F-actin severing protein GSNL-1/gelsolin (Meng et al., 2015). The phenotypes of gap junction mutants (*unc-1, unc-7, unc-9, unc-7 unc-9*) are different from those in CED or gsnl-1 mutants. Both CED/*gsnl-1*(lf) animals and control) animals at the L1 stage have similar number of SNB-1::GFP puncta at the end of D/V neurites. In contrast, gap junction mutant animals reported here already show differences of SNB-1::GFP from control animals in the L1 stage, that they have SNB-1::GFP puncta along the D/V neurites in addition to at the end of D/V neurites. When animals reach adult stages, CED or *gsnl-1(lf)* animals only have SNB-1::GFP at the end of D/V neurites, but gap junction mutants have SNB-1::GFP along the entire D/V neurites. While gap junction mutants have SNB-1::GFP distribution defects in axonal processes, no such phenotypes are observed in CED or *gsnl-1* mutants. Different from gap junction mutants, CED or *gsnl-1(lf)* animals have microtubule polarity in D/V neurites similar as that in control animals. With those differences and other evidence presented in this study, we conclude that GLR-RME gap junctions are involved in regulating RME axon specification rather than synapse elimination.

The authors will also need to have the manuscript undergo a substantial grammar check.

Thanks. We revised our manuscript to correct grammar and article usage.